# Bacmid Expression of Granulovirus Enhancin En3 Accumulates in Cell Soluble Fraction to Potentiate Nucleopolyhedrovirus Infection

**DOI:** 10.3390/v13071233

**Published:** 2021-06-25

**Authors:** Adriana Ricarte-Bermejo, Oihane Simón, Ana Beatriz Fernández, Trevor Williams, Primitivo Caballero

**Affiliations:** 1Institute for Multidisciplinary Research in Applied Biology, Universidad Pública de Navarra, 31006 Pamplona, Navarra, Spain; adriana.ricarte@unavarra.es (A.R.-B.); anabeatriz.fernandez@unavarra.es (A.B.F.); pcm92@unavarra.es (P.C.); 2Departamento de Investigación y Desarrollo, Bioinsectis SL, Polígono Industrial Mocholi Plaza Cein 5, Nave A14, 31110 Noain, Navarra, Spain; 3Instituto de Ecología AC, Xalapa, Veracruz 91073, Mexico; trevor.williams@inecol.mx

**Keywords:** *Alphabaculovirus*, *Betabaculovirus*, enhancins, AcMNPV recombinant, *Spodoptera exigua*, potentiation of pathogenicity

## Abstract

Enhancins are metalloproteinases that facilitate baculovirus infection in the insect midgut. They are more prevalent in granuloviruses (GVs), constituting up to 5% of the proteins of viral occlusion bodies (OBs). In nucleopolyhedroviruses (NPVs), in contrast, they are present in the envelope of the occlusion-derived virions (ODV). In the present study, we constructed a recombinant Autographa californica NPV (AcMNPV) that expressed the Trichoplusia ni GV (TnGV) enhancin 3 (En3), with the aim of increasing the presence of enhancin in the OBs or ODVs. En3 was successfully produced but did not localize to the OBs or the ODVs and accumulated in the soluble fraction of infected cells. As a result, increased OB pathogenicity was observed when OBs were administered in mixtures with the soluble fraction of infected cells. The mixture of OBs and the soluble fraction of Sf9 cells infected with BacPhEn3 recombinant virus was ~3- and ~4.7-fold more pathogenic than BacPh control OBs in the second and fourth instars of *Spodoptera exigua*, respectively. In contrast, when purified, recombinant BacPhEn3 OBs were as pathogenic as control BacPh OBs. The expression of En3 in the soluble fraction of insect cells may find applications in the development of virus-based insecticides with increased efficacy.

## 1. Introduction

Enhancins are virus-encoded metalloproteinases that can enhance the establishment of baculovirus infection by degrading the peritrophic matrix (PM) in the midgut of the host insect. They act as a protease and degrade the insect intestinal mucin (IIM), the major mucinous protein that constitutes the PM, resulting in the disruption of this structure and increasing its permeability to baculovirus occlusion-derived virions (ODVs) [1,2,3,4,5,6].

Enhancin genes are prevalent in lepidopteran-infecting granuloviruses (GV, genus *Betabaculovirus*), where they are present in the occlusion body (OB) matrix [5,6]. The virus-enhancing activity of betabaculovirus enhancins has been well-documented [6,7,8,9,10,11,12]. In contrast, orthologues present in lepidopteran-infecting nucleopolyhedroviruses (NPV, genus *Alphabaculovirus*) are associated with ODV envelopes and their efficacy as enhancers of NPV infections has come from experiments with recombinant viruses [6,13,14,15]. Some betabaculovirus and alphabaculovirus genomes contain multiple copies of these genes [13,14,15,16].

Most of the studies on infection potentiation by betabaculovirus enhancins have been performed using enhancins purified from OBs [7,8,9,10,12,17], whereas others have employed bacmid technology to express enhancin genes in recombinant NPVs [10,18,19,20,21]. One question concerning the recombinant enhancin expressed by an NPV is whether the recombinant enzyme is occluded within OBs, as in GVs, or integrated into the ODV envelope, as occurs in NPVs. Previous studies in which the Trichoplusia ni GV (TnGV) enhancin gene was expressed in recombinant Autographa californica NPV (AcMNPV) did not specifically address the localization of the recombinant protein [10,18,19]. For example, TnGV enhancin purified from cells infected by a recombinant AcMNPV effectively degraded PM proteins and increased lethal infection in host larvae [10]. Others have suggested that the enhancin may be occluded within OBs, as recombinant OBs were significantly more pathogenic than wild-type OBs in larval bioassays [18,19].

Interestingly, the incorporation of a betabaculovirus enhancin into alphabaculovirus OBs was achieved by fusing the enhancin gene to a C-terminal section of the polyhedrin gene [20,21]. Using this technology, a recombinant AcMNPV bacmid was constructed that embedded both Agrotis segetum GV enhancin and the Cydia pomonella GV GP37 enhancing factor within the OBs [20]. The GP37 enhancing factor is homologous to fusolin found in entomopoxviruses that binds chitin and enhances viral infection [22]. These recombinant viruses produced normal OBs that were significantly more pathogenic to susceptible larvae than wild-type OBs [20,21]. It is unclear, however, whether enhancin can be occluded into OBs or enveloped in the ODVs of an NPV without the use of the polyhedrin fusion technique.

The main objective of the present study was to determine whether an unmodified betabaculovirus enhancin, not fused with polyhedrin, was incorporated within NPV OBs or enveloped in the ODVs. For this, a polyhedrin-positive recombinant AcMNPV expressing a betabaculovirus enhancin was constructed. The localization of the recombinant protein was investigated by SDS-PAGE analyses and the insecticidal activity of the recombinant OBs was compared with that of wild-type OBs.

## 2. Materials and Methods

### 2.1. Insects, Cells and Viruses

Larvae of *Spodoptera exigua* were obtained from a healthy laboratory colony maintained on a semi-synthetic diet [23] at 25 ± 1 °C. Sf9 cells (ThermoFisher Scientific, Waltham, MA, USA) were maintained in TC100 medium (Lonza Bioscience, Cologne, Germany) supplemented with 10% fetal calf serum (Lonza Bioscience, Cologne, Germany) at 28 ± 1 °C [24]. For the construction of the recombinant virus expressing the enhancin gene, the Bac-to-Bac^TM^ Baculovirus Expression System was used (ThermoFisher Scientific, Waltham, MA, USA). The enhancin gene was amplified from Trichoplusia ni granulovirus (TnGV) [16], whereas the polyhedrin gene was amplified from the AcMNPV C6 clone, the type species of the *Alphabaculovirus* genus [25]. All viruses were obtained from the virus collection of the Microbial Bioinsecticides group at the Universidad Pública de Navarra.

### 2.2. Recombinant BacPh and BacPhEn3 Virus DNAs

We selected enhancin 3 (En3) of TnGV to construct the recombinant virus as TnGV-purified enhancins produced the highest potentiation (discussed in Appendix B). Among the three enhancins present in the TnGV genome [16], En3 possesses both the zinc-binding domain of metalloproteinases and the mucin-binding domain (Appendix A) (explained in Appendix C).

The recombinant virus was constructed using a Bac-to-Bac recombination system (ThermoFisher Scientific, Waltham, MA, USA) [26]. Two primer sets were designed to amplify the full-length coding sequence (from ATG to stop) of the enhancin 3 gene (*en3*, viral enhancing factor 3) of TnGV (TnOrf-149; nt 153,610 to 156,315 in the TnGV genome; accession number KU752557 or NC_038375), and the polyhedrin gene (*ph*) of AcMNPV (AcOrf-8; nt 4520 to 5257 in the AcMNPV-C6 genome; accession number L22858 or NC_001623) (Table 1). We did not include a His-tag sequence in the C-terminal for protein purification and detection, in order to produce a natural enhancin and to determine its natural location. Previously, the use of an N-terminal His-tag for identification and purification purposes resulted in reduced stability and binding properties of tagged proteins [27], and compromised the mucin degradation activity of the tagged enzyme in the case of enhancin-like Bel proteins from bacteria [28]. The genes were cloned in the pFastBacTM Dual (pFBD-phx-p10x) expression vector under the polyhedrin (*en3* gene) and p10 (*ph* gene) promoters. For the insertion of the specific genes, two pairs of primers were designed to amplify the *en3* gene of TnGV and *ph* of AcMNPV. The *Xba*I and *Pst*I restriction sites were introduced near the 5′ termini of the forward and reverse primers of *en3*, respectively, for further cloning and to direct the transcription of the *en3* with the polyhedrin promoter. Additionally, downstream from the *Xba*I restriction site and upstream from the ATG codon, a 20 nt sequence was inserted that corresponded to the *en3* promoter region, in order to transcribe this gene under its homologous promoter [10]. Similarly, for the *ph* gene, the *Xho*I and *Nco*I restriction sites were introduced near the 5′ termini of the forward and reverse primers, respectively, to direct cloning of the p10 promoter. The forward primer (Ph-Fw) was designed 308 nt upstream of the ATG codon, to avoid an AT rich region (69% AT and 31% GC), as baculovirus intergenic regions are AT rich and the primer could otherwise anneal at different points across the genome.

PCR amplifications were conducted using the Phusion high-fidelity Pfu DNA polymerase (New England Biolabs, Ipswich, MA, USA) and amplicons were recovered by using the PCR clean-up extraction kit NucleoSpin^®^ Extract II Kit (Macherey-Nagel, Düren, Germany). The *en3*- and *ph*-purified products were then ligated into pJET1.2/blunt plasmid (CloneJET PCR Cloning Kit, ThermoFisher Scientific, Waltham, MA, USA) following the manufacturer’s instructions. Ligation mixtures were transformed into *Escherichia coli* XL1-Blue electrocompetent cells using standard procedures. Positive clones were identified by colony-PCR. Five positive clones were grown, and plasmid DNAs were purified using the NucleoSpin R Plasmid Kit (Macherey-Nagel Inc., Düren, Germany). Subsequently, two selected plasmids for each gene were sequenced by ABI PRISM Big Dye Terminator Cycle Sequencing Ready Reaction Kits and an ABI PRISM 3100 Genetic Analyzer (STABVida, Caparica, Portugal). After sequence confirmation, a selected clone was digested with *Xba*I and *Pst*I enzymes and electrophoresed in 1% agarose gel. The generated restriction fragment was extracted from the gel and purified by column and ligated into *Xba*I-*Pst*I-digested pFBD vector to generate the pFBD-En3. Two independent clones were selected for sequencing as described above. Once the correct insertion of *en3* was confirmed, the *ph* gene was cloned in the pFBD-En3 by double digestion with *Xho*I-*Nco*I to obtain the recombinant plasmid pFBD-PhEn3, under the control of the p10 promoter. Two independent clones were selected for sequencing to confirm the correct insertion of the *ph* gene. Finally, ligation products were electroporated into *E. coli* DH10BacTM cells (ThermoFisher Scientific, Waltham, MA, USA) that contained the AcMNPV shuttle vector (bacmid) and the helper plasmid, to produce the recombinant bacmid BacPhEn3, following transposition of the pFBD-PhEn3 expression construct through the transposition of site-specific cassettes between Tn7R and Tn7L and the subsequent steps following standard protocols. In parallel, a negative control virus was constructed that included the *ph* gene without any gene of interest, to generate the pFBD-Ph vector, previously digested with *Xho*I-*Nco*I enzymes, under the *p10* promoter, following the same procedure.

Colonies were selected on LB (Luria Bertani) agar plates containing kanamycin (50 µg/mL), gentamycin (7 µg/mL), tetracycline (10 µg/mL), X-gal (100 µg/mL) and IPTG (40 µg/mL). Ten white colonies were re-streaked on fresh LB agar plates under the same conditions to avoid contamination. Recombinant bacmid DNA was isolated using the PureLinkTM HiPure Plasmid DNA Miniprep Kit (ThermoFisher Scientific, Waltham, MA, USA). The successful transposition to bacmid was confirmed by digestion of DNA with *Pst*I and by PCR amplification and sequence analysis using M13-Fw and M13-Rv primers that annealed outside the coding region of the *p10* and *ph* promoters in the pFBD vector (Table 1).

### 2.3. Transfection of Sf9 Cells with Recombinant BacPh and BacPhEn3 Virus DNAs

Recombinant baculoviruses were produced in a 25 cm^2^ cell culture flask by transfecting 1 μg of BacPhEn3 and BacPh genomic DNAs into 10^6^ Sf9 cells using Lipofectin^®^ Reagent (ThermoFisher Scientific, Waltham, MA, USA) [24,29]. The resulting OBs of both recombinant viruses were compared with those produced by the wild-type AcMNPV C6. For this, 10^6^ Sf9 cells were inoculated with AcMNPV-C6 budded virions (BVs) at a m.o.i. of 10 that we had in stock. The production of OBs in cells was checked daily, and five days post-infection, the supernatant containing BVs and the pellet containing the cells and viral OBs were recovered by centrifugation (2400× *g*, 5 min). DNA extraction was performed on infected cells [24] to confirm the restriction endonuclease (REN) profiles of the recombinant bacmids and to check for cross-contamination. Finally, to evaluate the normal occlusion of ODVs within the OBs of the two recombinants, the number of infectious units within OBs was compared with that of wild-type virus AcMNPV-C6 by end-point dilution in Sf9 cells [24,29]. Specifically, at five days post-infection, OBs were released from cells by sonication and OBs were counted in triplicate samples in a Neubauer hemocytometer. A 500 µL volume of suspension containing 10^8^ OBs/mL was mixed with an equal volume of 0.1 M Na_2_CO_3_ and stirred at 28 °C for 30 min. Undissolved OBs and cell debris were removed by centrifugation (5900× *g*, 5 min), and the supernatant containing the ODVs was filtered through a 0.45 µm filter and used in the end-point dilution assay [23,26].

### 2.4. Production of BacPh and BacEn3 OBs in Larvae

The BacPh and BacPhEn OBs were produced in larvae by injecting the BV suspension obtained from transfection assays. For this, BV suspensions were initially quantified by plaque assay [24,29] and both suspensions had similar titers (3.51 × 10^7^ pfu/mL for BacPh and 3.80 × 10^7^ pfu/mL for BacPhEn3; *t* = 2.139; d.f. = 6; *p* = 0.076). Therefore, recombinant bacmid OBs were amplified by injecting the same quantity of BVs of BacPh and BacPhEn3, 8 μL of BV suspension (1:1000), into *S. exigua* fifth instar larvae from the laboratory colony. Groups of 25 larvae were inoculated with each virus and were reared individually in 30 mL plastic cups with a piece of semi-synthetic diet. Virus-killed larvae were collected daily and transferred to a 50 mL collection tube. OBs were purified from dead larvae by filtration through muslin and several rounds of centrifugation in 0.1% (wt/vol) sodium dodecyl sulfate (SDS) at 2400× *g* 5 min. Finally, OBs were resuspended in 1 mL double-distilled water and stored at 4 °C until required. The fidelity of OBs produced in insects was confirmed by REN analysis and by sequencing of the PCR products obtained following amplification using M13-Fw and M13-Rv primers (Table 1).

### 2.5. Detection and Localization of En3 Protein

To detect and localize the TnGV En3 protein an SDS-PAGE was performed using different samples of cells and OBs produced both in cell culture and in larvae. The protocol to obtain the different samples was as follows. A 25 cm^2^ cell culture flask containing a batch of 10^6^ Sf9 cells was transfected with the DNAs of BacPh and BacPhEn3, as mentioned in Section 2.3. At five days post-infection, the Sf9 cells and medium were recovered with a cell scraper (Bio-Rad, Berkeley, CA, USA). To separate the medium containing the BVs from the cells containing the OBs, a low-speed centrifugation was performed at 2400× *g* during 5 min. The supernatant sample containing the BVs was transferred to a 15 mL collection tube (Sample 1; S1). The pellet with the cells and OBs were rinsed twice with 1 mL phosphatase-buffered saline (PBS) pH 7.4 and resuspended in 1 mL PBS. OBs were released from cells by sonication in an ultrasonic bath (Selecta Master, JP Selecta, Abrera, Spain) for 10 min at maximum power (100 W) to produce sample 2 (S2). The lysate was then centrifuged at 5900× *g* for 5 min. The pellet containing the cell debris and OBs was rinsed twice with 500 µL of PBS and resuspended in 500 µL PBS to produce sample 3 (S3). The supernatant containing the soluble phase of the infected cells was transferred to a 1.5 mL microcentrifuge tube to produce sample 4 (S4). The OBs produced in larvae and semi-purified by filtration through muslin and differential centrifugation (2400× *g* during 5 min) were also analyzed as sample 5 (S5). Three replicates were performed in parallel.

For each sample (S1–S5) and virus, an aliquot of 20 µL was mixed with the same volume of 2× sample buffer (Bio-Rad, Berkeley, CA, USA), boiled at 100 °C for 5 min, and then subjected to electrophoresis as previously described [30], using Criterion TGX™ 4–20% Precast Gel (Bio-Rad, Berkeley, CA, USA). Gels were stained with Coomassie Brilliant Blue R-250 (Bio-Rad, Berkely, CA, USA) and then distained in 30% ethanol and 10% acetic acid and photographed.

### 2.6. Insect Bioassays

To determine the insecticidal characteristics and the enhancement activity of BacPolhEn3 recombinant OBs, insect bioassays were performed in second and fourth instar larvae of *S. exigua*. OBs originated from lysate cell culture and the OBs were produced in larvae. OBs from lysate cultures were used directly without any purification (OBs and cell medium), whereas those produced in larvae were purified by filtration and several rounds of centrifugation. Triplicate samples of OBs were counted in a Neubauer hemocytometer.

A discriminating concentration assay was initially performed using two concentrations of OBs for each instar. Second instars were inoculated with 2 × 10^4^ and 2 × 10^5^ OBs/mL, whereas fourth instars were inoculated with 10^5^ and 10^6^ OBs/mL using the droplet feeding method [31]. Larvae were starved for 8 to 12 h at 25 ± 1 °C and then allowed to drink from an aqueous suspension of OBs and 10% sucrose plus 0.001% (*w*/*v*) Fluorella Blue. Control larvae were treated identically but did not consume OBs. Groups of 28 larvae that ingested the suspension within 10 min were transferred individually to the wells of a 28-well plate with a piece of semi-synthetic formaldehyde-free diet [23]. Larvae were reared at 26 ± 1 °C and mortality was recorded daily until insects had either died or pupated. Bioassays were performed five times (replicates) using different batches of larvae. Virus-induced mortality results were analyzed within each instar and each OB concentration, to compare the OBs from cell culture lysate with those purified from larvae. A Shapiro–Wilk test and Levene’s test indicated that the data were normally distributed with homogeneity of variance. The results were then subjected to three-way analysis of variance (ANOVA) with virus (BacPh and BacPhEn3), origin (OBs + cell lysate and OBs from larva), and inoculum concentration (high and low) specified as factors. Means were compared by Tukey test. The analysis was performed in the R-based package Jamovi v.1.2.27.0 [32].

The pathogenicity of BacPhEn3 OBs produced in cell culture and in larvae was compared with that of BacPh OBs in concentration-mortality bioassays in second and fourth instars of *S. exigua*. Bioassays were performed following the droplet feeding method [31] to inoculate larvae with one of five concentrations of OBs. For second instars, the concentrations were 2 × 10^7^, 4 × 10^6^, 8 × 10^5^, 1.6 × 10^5^ and 3.2 × 10^4^ OBs/mL, whereas for fourth instars, 1 × 10^8^, 1 × 10^7^, 1 × 10^6^, 1 × 10^5^ and 1 × 10^4^ OBs/mL were used to inoculate larvae. These concentrations were previously determined to kill between 95% and 5% of the experimental insects. Control larvae were treated identically but did not consume OBs. Groups of 28 larvae that ingested the suspension within 10 min were reared individually at 26 ± 1 °C and mortality was recorded daily until insects had either died or pupated. The bioassay was performed three times using different batches of larvae. Virus-induced mortality results were subjected to probit analysis using the Polo-Plus program [33].

## 3. Results

### 3.1. Recombinant BacPh and BacPhEn3 Virus DNAs

The *polyhedrin* and *enhancin 3* genes were amplified by PCR from AcMNPV C6 and TnGV DNAs, respectively, to obtain fragments of 1073 bp and 2738 bp (data not shown). Recombinant AcMNPV viruses that included the *ph* gene (BacPh), and the *ph* and *en3* genes (BacPhEn3) were constructed using Bac-to-Bac technology. Correct insertion of these genes was confirmed by restriction endonuclease (REN) analysis (Figure 1a) and sequencing analysis following PCR amplification of adjacent regions using the M13-Fw and M13-Rv primers (Figure 1b).

PCR amplicons of ~3.5 kb and ~6.5 kb for BacPh and BacPhEn3, respectively, were in line with the expected sizes of 3633 and 6371 bp obtained after adding 2560 bp to each of the PCR products following the Bac-to-Bac protocol (ThermoFisher Scientific, Waltham, MA, USA). In fact, the *en3* was a 2738 bp fragment, so the fragment obtained with BacPhEn3 DNA was ~3 kb larger in the gel. Sequence analysis confirmed that the *ph* gene in BacPh and BacPhEn3 recombinants and the *en3* in the BacPhEn3 recombinant were inserted correctly after the *p10* promoter and *ph* promoter, respectively. DNA was then purified and transfected into Sf9 cells for the production of viral OBs.

### 3.2. Transfection of Sf9 Cells with Recombinant BacPh and BacPhEn3 Virus DNAs

At five days after transfection, most of the cells infected with BacPolhEn3 had OBs in the cell nuclei at levels similar to those of the control virus (BacPolh) and AcMNPV C6. Recombinant viruses produced normal OBs observable by optical microscopy (Figure 2a). The shape and size of OBs were indistinguishable among the three viruses (Figure 2a) and the ODV content (number of infectious units) of samples comprising 5 × 10^8^ OBs did not differ significantly among these viruses (ANOVA, *F*_2,6_ = 0.919; *p* = 0.449) (Figure 2b).

Examination of restriction endonuclease profiles and PCR analysis of recombinant OBs was performed to corroborate their identity in comparison with the original samples. Infected cells were recovered and lysed to obtain viral OBs, and DNA extraction was performed on these OBs. REN analyses showed that viral DNA profiles from BacPolh and BacPolhEn3 transfections (Figure 3a) were the same as those of the original inocula (shown in Figure 1a). M13-Fw and M13-Rv primers produced a PCR product of ~3.5 kb and ~6.5 kb for BacPh and BacPhEn3, respectively, which corresponded with the expected sizes (Figure 1b). Finally, sequence analysis confirmed the presence of these genes in the correct position, and that the recombinant viruses were correctly designed and constructed (data not shown).

### 3.3. Production of BacPh and BacPhEn3 OBs in Larvae

The supernatant containing BVs was used to inject *S. exigua* fifth instar larvae to produce large quantities of OBs. Injection of BacPh and BacPhEn3 BVs resulted in a similar prevalence of larvae mortality (range 86–95%). Dead larvae were collected five to seven days after infection and showed the typical signs of lethal polyhedrosis disease. OBs were purified and were checked by REN and PCR analysis. REN treatment of BacPh and BacPhEn3 OBs resulted in the characteristic profiles of each virus (Figure 3b), identical to those of the original inoculated viruses (shown in Figure 1a). Sequence analysis of PCR products confirmed that sequences were identical to those of the original inocula. Therefore, the BacPh and BacPhEn3 OBs produced in larvae were confirmed to be those of the recombinant viruses and were used for bioassays.

### 3.4. Enhancin 3 Is Solubilized in the Cell Medium

To detect En3, a sample separation protocol was followed (Figure 4a), and an aliquot of each sample was used to perform SDS-PAGE (Figure 4b).

The En3 protein was present in Sf9 cells transfected with the recombinant BacPhEn3 but was not present in BacPh transfected cells. The SDS-PAGE analysis revealed that the BV fraction (S1) presented a major band of 56.8 kDa, corresponding to the GP64 protein (indicated by the letter “a” in both BacPh and BacPhEn3; Figure 4b), and lacked the band corresponding to En3. Cell pellets resulting from infection with BacPhEn3 showed a band of ~100 kDa that corresponded in size to En3 (indicated by the letter “c” in S2 in BacPhEn3; Figure 4b), which was absent in the S2 sample of BacPh. A band of similar size to En3 was present in the S2 sample of BacPh, but this had a lower molecular weight and also appeared below the En3 band in S2 of BacPhEn3 (indicated with an asterisk in S2 in both BacPh and BacPhEn3). Moreover, in S4 of BacPhEn3, only the upper band (labeled “a”) was present, corresponding to En3, which co-migrated with En3 in S2 of BacPhEn3 (also labeled “a”), rather than the lower band (labeled “*”) in S2 of BacPhEn3.

Cell pellets were lysed by sonication and the pellet, and the supernatant obtained after low-speed centrifugation showed the band of ~100 kDa that corresponded in size to En3 (indicated with the letter “c” in S4 in BacPhEn3; Figure 4b). As a control, cells infected with BacPh did not show the band corresponding to En3 in the supernatant of cell lysate (S4 sample in BacPh; Figure 4b). The purified OBs from lysed cells (S3) did not show the band corresponding to En3, indicating that this protein was not present in the OBs or ODVs at detectable quantities (S3 sample in BacPhEn3; Figure 4b). A band corresponding to En3 was not observed in the OBs purified from larvae and was therefore also not present in the ODVs at detectable levels (S5 sample; Figure 4c).

Finally, the polyhedrin protein of 27 kDa was present in cell pellets and OBs produced in larvae infected by BacPh and BacPhEn3 (S2 and S3; Figure 4b and S5; Figure 4c), indicating the correct production of this OB protein, as expected. In the S4 samples of BacPh and BacPhEn3, a faint band of similar size to Polh was present (indicated with a cross in S4) but was clearly less intense than in S2 and S3. To separate the OBs (S3) and the soluble fraction of the cells (S4), a low-speed centrifugation was performed; therefore, the soluble fraction (S4) may have contained traces of OBs. It is also possible that this band could be another cellular protein, as it also appeared in the S1 sample (BV fraction) in both BacPh and BacPhEn3.

These results clearly indicated that En3 was produced but was not incorporated into OBs or enveloped within ODVs, but instead accumulated in the cell medium as a soluble entity.

### 3.5. Biological Activity of BacPhEn3 OBs Produced in Cell Culture and in Larvae

Discriminating concentration insect bioassays were performed with the OBs and cell lysate from cell culture (equivalent to S2 in Figure 4a) and with OBs purified from larvae (equivalent to S5 in Figure 4a). In second instars (Figure 5a), a significant interaction effect was detected in virus × origin (cells or larvae) of OBs, presumably due to the presence of En3 in the cell lysate of BacPhEn3-infected cells (*F*_1,32_ = 5.460; *p* = 0.026). The origin of the inoculum (cells vs. larvae) also interacted significantly with inoculum concentration (*F*_1,32_ = 6.982; *p* = 0.013). The main effects of virus (*F*_1,32_ = 0.882; *p* = 0.355) and inoculum origin (*F*_1,32_ = 0.058; *p* = 0.812) were not significant, whereas mortality increased significantly with OB concentration, as expected (*F*_1,32_ = 214.71; *p* < 0.001).

The results differed markedly in fourth instars compared to second instar larvae (Figure 5b). The main effects of virus (*F*_1,32_ = 21.20; *p* < 0.001), inoculum origin (*F*_1,32_ = 15.43; *p* < 0.001) and inoculum concentration (*F*_1,32_ = 122.54; *p* < 0.001) significantly affected larval mortality. There was also a significant interaction of virus × origin of OBs (*F*_1,32_ = 11.572; *p* = 0.002), due to the presence of Eh3 in the cell lysate of BacPhEn3 OBs + cell lysate, which resulted in the highest prevalence of mortality in the high inoculum concentration (65%, Figure 5b), and its presumed absence in BacPhEn3 OBs purified from larvae (39% mortality). The tendency was the same in the low inoculum concentration, although the mortality induced by BacPh OBs and BacPhEn3 OBs did not differ significantly when inoculated in mixtures with infected cell lysate (Tukey, *p* > 0.05). None of the other interaction terms were significant (*p* > 0.05).

As the discriminating concentration assays revealed that BacPhEn3 OBs + cell lysate resulted in high mortality in fourth instar larvae, a series of concentration-mortality response bioassays was performed. Specifically, the LC_50_ values and relative potencies of OBs were estimated for both recombinants obtained from cell culture and larvae based on the prevalence of virus-induced mortality observed across a range of five OB concentrations (Table 2).

In second instar larvae, the BacPhEn3 OBs + cell lysate were 2.95-fold more pathogenic than control BacPh OBs + cell lysate, presumably due to the presence of En3 in the lysate fraction. In contrast, BacPh and BacPhEn3 OBs purified from larvae were as pathogenic as the control virus with relative potency values of 1.16 and 0.65, respectively, with broadly overlapping 95% confidence limits, reflecting the absence of En3 in these samples.

The enhancement of infection of BacPhEn3 OBs + cell lysate was more marked in fourth instar larvae. The BacPhEn3 OBs + cell lysate was 4.69-fold more pathogenic than control BacPh OBs + cell lysate, whereas BacPh OBs and BacPhEn3 OBs purified from larvae were as pathogenic as the control BacPh OBs + cell lysate with a relative potency of 1.01 and 0.97, respectively. These findings confirmed that the lysate of BacPhEn3-infected cells was responsible for the potentiation of AcMNPV infection.

## 4. Discussion

The localization of a granulovirus enhancin in NPV-infected Sf9 cells and its enhancing activity was investigated by constructing an AcMNPV recombinant that expressed the enhancin 3 (*en3*) gene of TnGV. The enhancin protein was localized to the soluble fraction of the cells and was not present in NPV OBs (or ODVs) in detectable quantities. Following cell lysis, the enhancing factor present in the lysate was capable of potentiating NPV infection by ~3-fold in second instars and ~4.7-fold in the fourth instar larvae of *S. exigua*.

These findings are consistent with the purification of enhancin from *T. ni* cells infected with recombinant AcMNPV that expressed the TnGV En3 reported by Lepore et al. [10]. In contrast, Hayakawa et al. [18] used intact virus infected Sf9 cells as inoculum. These cells were infected with a polyhedrin-negative AcMNPV recombinant virus that expressed the En3 of TnGV, so that larvae in the bioassay ingested mixtures of wild-type AcMNPV OBs and cells infected by the recombinant virus that likely contained the enhancin protein. In another study, recombinant virus expressing the TnGV enhancin (En3) was subjected to three rounds of passage in cell culture prior to insect bioassay. The resulting OBs were observed to have modest levels (two-fold) of potentiation activity in a diet contamination bioassay, although the purification steps of the OBs were not described [19]. We assume that the OB preparations used by these authors included traces of enhancin from the host cells, which resulted in the modest levels of potentiation observed.

In the present study, we demonstrated that the lysate of infected cells contained the potentiation factor rather than the recombinant OBs or ODVs. The BacPhEn3 OBs purified from infected larvae by filtration and differential centrifugation were as pathogenic as control BacPh OBs purified from larvae or BacPh OBs inoculated in mixtures with cell lysate. Potentiation activity was only observed for BacPhEn3 OBs when in mixtures with the lysate of BacPhEn3-infected cells. The prevalence of virus-induced mortality could have been influenced by the presence of non-occluded virions in cells. If so, such entities would be present in both control (BacPh) and BacPhEn3 treatments involving cell lysate. In addition, the BV and OB titers obtained in infected cells were similar for the two viruses, suggesting similar titers of non-occluded virions in both BacPh and BacPhEn3 treatments. Therefore, we can conclude that the potentiation effect was due to the presence of En3 in the lysate of BacPhEn3-infected cells.

When purifying OBs from virus-killed insects, once the infected larvae had liquefied, they were homogenized and filtered to eliminate debris. The resulting suspension was subjected to various rounds of low-speed centrifugation to pellet the OBs, whereas the supernatant containing traces of cell debris and cellular proteins was discarded to finally obtain semi-purified OBs [34,35]. Therefore, any enhancing factor was eliminated or markedly reduced in concentration during OB purification from larvae. It is possible that bioassay results could have been influenced by the presence of non-occluded virions in cell lysates. However, if present, such entities would be present in similar quantities in both control (BacPh) and BacPhEn3 treatments involving cell lysate, so we are confident that the effects attributed to the presence of enhancin are real and not an inadvertent effect of the presence of virions that had not been incorporated into OBs.

The enhancing activity demonstrated in this study (~3-fold and ~4.7-fold in second and fourth instars, respectively) was broadly in line with that reported by other authors [10,19], although Hayakawa et al. [18] reported higher potentiation of AcMNPV and SeMNPV infection in *S. exigua* third instars, for the reasons mentioned above. In other systems, a recombinant AcMNPV expressing the enhancing-like protein of AgseGV was fivefold more pathogenic than the wild-type OBs [36]. Using a fusion technology, the AgseGV enhancin and GP37 were embedded independently into two different recombinants, and the recombinant bacmid OBs were three- and fivefold more pathogenic than wild-type OBs, respectively [21]. The GP37 is homologous to the glycoprotein fusolin present in entomopoxvirus spindles, the N-terminal of which binds to chitin and markedly enhances infection by entomopoxviruses and nucleopolyhedroviruses [22]. Similarly, using the same technology, both AgseGV enhancin and GP37 were embedded in the same recombinant OBs that were 3.9-fold and 7.4-fold more pathogenic to second and fourth instar larvae than wild-type OBs, respectively [20].

We observed a clear instar-dependent effect in the potentiation activity of BacPhEn3 cell lysate. Enhancin potentiation tends to be greater in late instars that are normally more resistant to baculovirus infection than their younger counterparts [4,12,20]. The lepidopteran PM contains pores that vary in size among the different species and across larval instars [4]. In early instars, the PM is generally less tightly constructed and more permeable, whereas in later instars, the PM is well-formed and has lower porosity [4,6,37]. Therefore, PM disruption by these enzymes is greater in late instars and the potentiation effect more marked as late instar larvae are usually more resistant to baculovirus infection.

Enhancin orthologues are present in a few group II alphabaculoviruses [13,14,15,38,39]. These proteins are present within ODVs in association with nucleocapsids, where they appear to facilitate access to the midgut cells [15,40]. They also target IIM, with the mucins being common targets for both GV and NPV enhancins [40,41]. Indeed, GV and NPV enhancins are similar but differ in several aspects. They share up to 39% identity, range in size from 758 to 848 aa, and contain a conserved zinc-binding domain that is characteristic of metalloproteases [6,42]. However, NPV enhancins contain a potential transmembrane domain followed by a series of basic amino acid residues at their carboxyl terminus (Appendix A) [6]. The presence of this transmembrane domain allows them act as fusains targeted at midgut cells [6]. The absence of the transmembrane domain in granulovirus enhancins would prevent the envelopment of TnGV En3 in the AcMNPV ODV membrane. In contrast, NPV enhancins are able to insert into the ODV membrane of AcMNPV. For example, the Mamestra configurata NPV enhancin was found to be a component of the ODVs of a recombinant AcMNPV expressing this protein [40], and the recombinant OBs were over fourfold more pathogenic than wild-type OBs [14].

As GV and NPV enhancins clearly have different functions, the incorporation of a GV-type enhancin within alphabaculovirus OBs would appear challenging. This was recently achieved by fusion of the enhancin gene with the polyhedrin gene [20,21,43]. The application of gene fusion technology has the potential to create a novel protein expression system and an efficient virus-based system for insecticide production in countries that allow the use of genetically modified organisms in agriculture [44]. However, the use of recombinant viruses in agriculture is currently prohibited in many countries including those of the European Union (Directive 2001/18/CE).

## 5. Conclusions

In the present study, a granulovirus enhancin produced by an AcMNPV recombinant accumulated within infected cells and was shown to be present in the soluble fraction of the cells following lysis. The soluble fraction was responsible for potentiation of AcMNPV infection in *S. exigua* larvae and was more effective in fourth instars than in second instar larvae. The production of solubilized enhancins using a baculovirus-based expression systems could be used to improve the efficacy of biological insecticides against lepidopteran pests.

## Figures and Tables

**Figure 1 viruses-13-01233-f001:**
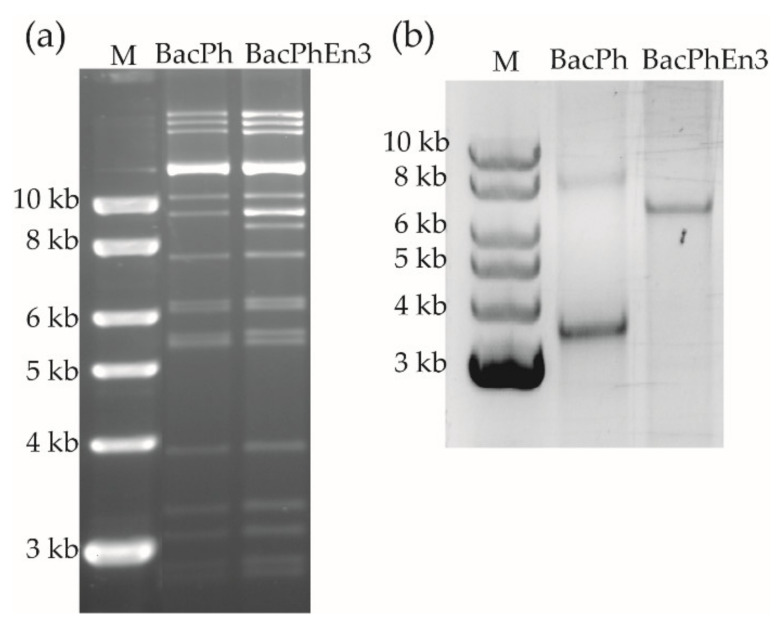
(**a**) *Pst*I restriction endonuclease profiles of recombinant bacmids BacPh and BacPhEn3. The strong band above the 10 Kb marker corresponds to the Bac-to-Bac helper plasmid. (**b**) Inverted image of PCR products obtained after amplification of recombinant DNAs using M13-Fw and M13-Rv primers. The image was inverted for improved clarity.

**Figure 2 viruses-13-01233-f002:**
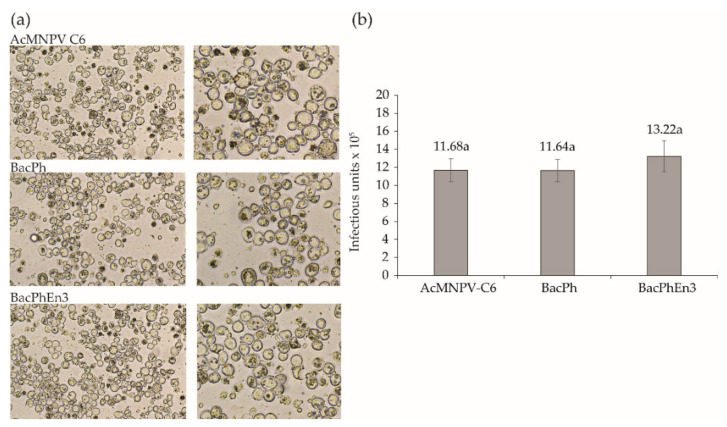
(**a**) Sf9 cells infected with AcMNPV-C6 BVs at 10 m.o.i., and Sf9 cells transfected with recombinant BacPh and BacPhEn3. The images were taken 5 days after infection; on the left, low magnification (40×) images of infected Sf9 cells and on the right, higher magnification (100×) images of cells presenting normal presence and abundance of OBs in cell nuclei. (**b**) Results of end-point dilution assay of the ODVs released from samples of 5 × 10^8^ OBs obtained from AcMNPV-C6 OBs and cell pellets of infected BacPh and BacPhEn3 cells showing similar numbers of infectious units (ANOVA, *p* > 0.05). Values above columns indicate means. Error bars indicate standard deviation.

**Figure 3 viruses-13-01233-f003:**
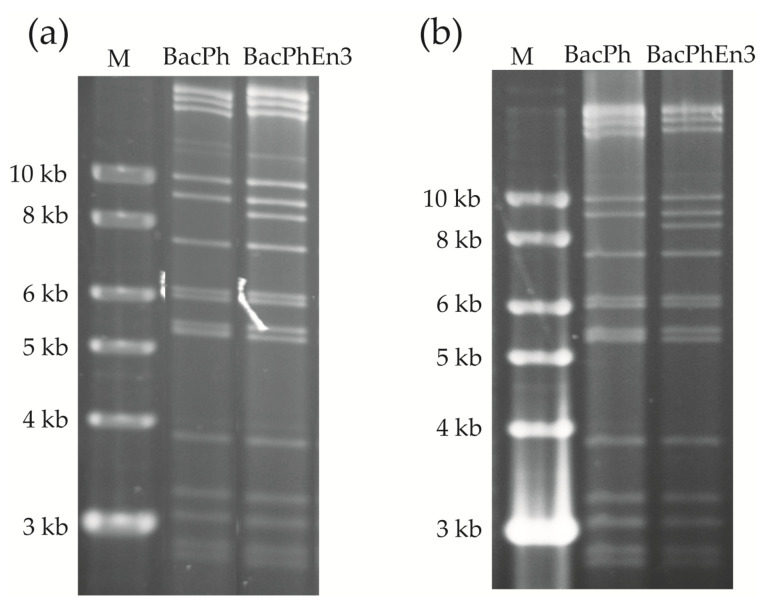
Restriction endonuclease profiles with *Pst*I of (**a**) cell pellets transfected with recombinant bacmids BacPh and BacPhEn3, and (**b**) OBs purified from larvae after injection of transfection supernatant (BVs) in *Spodoptera exigua* fifth instars. The molecular size marker, M, was smart ladder (Stratagene).

**Figure 4 viruses-13-01233-f004:**
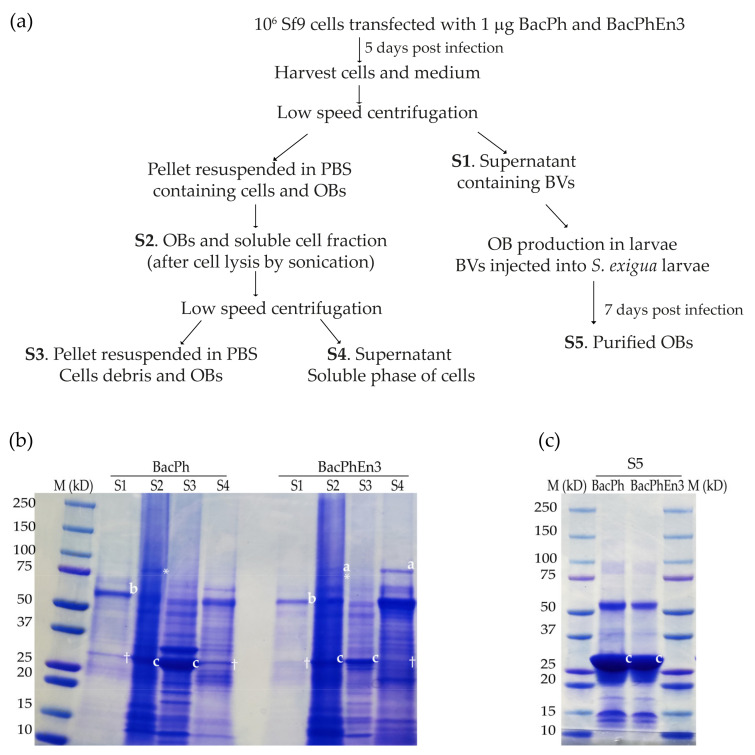
(**a**) Sample extraction protocol for virus-infected cells and OBs resulting in five types of samples (S1–S5). For each sample, a 20 µL sample was subjected to SDS-PAGE. (**b**) SDS-PAGE of samples S1–S4 obtained from cell culture. Letters on gel image indicate (**a**) the En3 protein in S2 and S4 in BacPhEn3, (**b**) GP64 in S1 for both BacPh and BacPhEn3 and (**c**) the polyhedrin in S2, S3 and S5, for both BacPh and BacPhEn3. Signs on gel images indicate (†) the band similar to polyhedrin and (*) the band immediately below the En3 protein in S2 in both BacPh and BacPhEn3. (**c**) SDS-PAGE of sample S5 purified from virus-killed larvae shown in (**a**). Letters on gel image indicate (**c**) the polyhedrin in S2, S3 and S5, for both BacPh and BacPhEn3.

**Figure 5 viruses-13-01233-f005:**
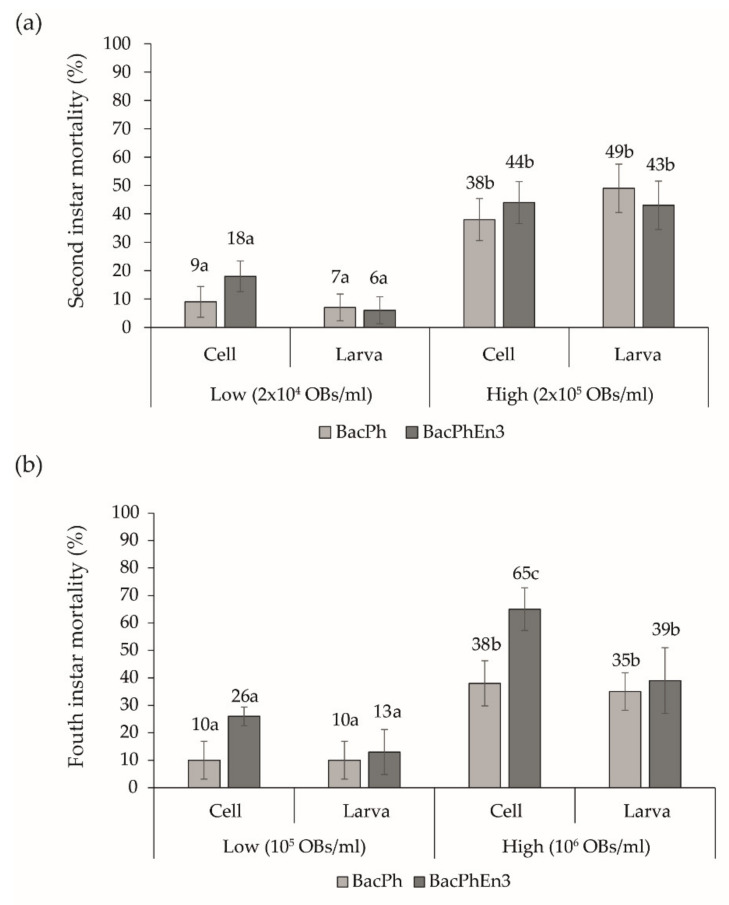
Percentage of larval mortality induced by BacPh and BacPhEn3 OBs produced in cell culture (OBs + cell lysate) and OBs purified from larvae in (**a**) second instars and (**b**) fourth instars of *Spodoptera exigua*. Second instars were inoculated with 2 × 10^4^ OBs/mL or 2 × 10^5^ OBs/mL and fourth instars with 10^5^ OBs/mL or 10^6^ OBs/mL. Values above columns indicate mean percentage of mortality. Values followed by identical letters did not differ significantly (ANOVA, Tukey *p* < 0.05). Error bars indicate SD.

**Table 1 viruses-13-01233-t001:** Primers used in this study.

Primer	Sequence	Amplification Purpose
En3-TnGV-Fw	5′-TCTCTAGAGCTGCATTAATTATAAGACTATGTC -3	En3 amplification. TnGV DNA was used a template. Underlined—the *Xba*I site. Double underlined—En3 promoter and ATG start codon; nt 153,590 to 153,614 in the TnGV genome. Tm 51 °C.
En3-TnGV-Rv	5′-CCCTGCAGTTAGAACGCTATCATTTTTAACG-3	En3 amplification. TnGV DNA was used a template. Underlined—the *Pst*I site. Double underlined —the *en3* TAA stop codon; nt 156,293 to 156,315 in TnGV genome. Tm 50 °C.
Ph-Fw	5′-CGCTCGAGGCCGGCATAGTACGC-3′	Ph amplification. AcMNPV C6 DNA was used a template. Underlined—the *Xho*I site; nt 4197 to 4211 in AcMNPV genome. Tm 56 °C.
Ph-Rv	5′-CGCCATGGTTAATACGCCGGACCAGTG-3′	Ph amplification. AcMNPV C6 DNA was used a template. Underlined—the *Nco*I site. Double underlined—the *ph* TAA stop codon; nt 5239 to 5257 in AcMNPV genome. Tm 54 °C.
En3-Seq-Fw	5′-CCGTACCCGCAAATATG-3′	En3 sequence confirmation. Primer that annealed at nt 1053 to 1073 in the *en3* gene. Tm 53 °C.
pFBD-seq-Fw	5′-CCGTGTTTCAGTTAGCC-3′	En3 sequence confirmation. Primer that annealed at nt 7563 to 7579 in pFBD plasmid. Tm 54 °C.
M13-Fw	5’-CCCAGTCACGACGTTGTAAAACG-3´	For confirmation of correct insertion of *en3* and *ph*. Primer that flanked the mini att-Tn7 site of the bacmid. Tm 54 °C.
M13-Rv	5´-AGCGGATAACAATTTCACACAGG-3´	For confirmation of correct insertion of *en3* and *ph*. Primer that flanked the mini att-Tn7 site of the bacmid. Tm 53 °C.

**Table 2 viruses-13-01233-t002:** OB pathogenicity (LC_50_) in *Spodoptera exigua* second and fourth instars inoculated with BacPh and BacPhEn3 OBs + cell lysate from infected Sf9 cells and BacPh and BacPhEn3 OBs purified from larvae following injection with BacPh and BacPhEn3 BVs.

Instar	Virus	Slope ± SE	LC_50_ (OBs/mL)	95% Conf. Interval		Relative	95% Conf. Interval	
				Lower	Upper	Potency	Lower	Upper
Second instar	OBs and cell lysate from Sf9 cells							
	BacPh	0.306 ± 0.078	1.33 × 10^5^	4.30 × 10^4^	8.75 × 10^5^	1		
	BacPhEn3	0.573 ± 0.084	4.51 × 10^4^	2.37 × 10^4^	8.73 × 10^4^	2.95	1.03	11.40
	OBs purified from larvae following injection with BVs							
	BacPh	0.555 ± 0.088	1.15 × 10^5^	5.93 × 10^4^	2.58 × 10^5^	1.16	0.27	5.03
	BacPhEn3	0.488 ± 0.093	2.03 × 10^5^	8.73 × 10^4^	7.52 × 10^5^	0.65	0.13	3.33
Fourth instar	OBs and cell lysate from Sf9 cells							
	BacPh	0.847 ± 0.126	1.10 × 10^6^	6.15 × 10^5^	2.84 × 10^6^	1	-	-
	BacPhEn3	1.019 ± 0.119	2.34 × 10^5^	1.66 × 10^5^	3.53 × 10^5^	4.69	2.63	10.59
	OBs purified from larvae following injection with BVs							
	BacPh	0.830 ± 0.110	1.09 × 10^6^	6.38 × 10^5^	2.14 × 10^6^	1.01	0.40	2.07
	BacPhEn3	0.756 ± 0.106	1.13 × 10^6^	6.32 × 10^5^	2.37 × 10^6^	0.97	0.37	2.02

Relative potency values were calculated with respect to the BacPh OBs + cell lysate (control) treatment in both instars. Goodness-of-fit tests were non-significant for second (*χ*^2^ = 6.82; df = 3; *p* = 0.078) and fourth (*χ*^2^ = 2.85; df = 3; *p* = 0.416) instars.

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
