# Peer review of "Bacmid Expression of Granulovirus Enhancin En3 Accumulates in Cell Soluble Fraction to Potentiate Nucleopolyhedrovirus Infection"

_viruses, 2021, doi:10.3390/v13071233_

Round 1

Reviewer 1 Report

Dear authors

The manuscript was well written and presented interesting results which further expand upon the application of baculovirus enhancins as a means of improving viral infection. I have two comments, one relating to band identification in figure 4 and another relating to further experimentation using the BacPhEn3 S4 sample which may further support the presence of en3 in the soluble fraction and it's role in enhancing infection. Additional minor comments are provided below.

Comments:

My first comment relates to Figure 4 and how the various proteins were identified. It would appear bands were selected based on estimated sizes of the target proteins rather than via any form of purification or specific detection (such as western blot). Was there some form of analysis not mentioned in the methods section which aided in the identification of proteins? For example, the authors mark two bands in S2 and S4 of BacPhEn3 as enhancin, however, a similar sized protein band is visible in BacPh S2 when roughly based on the marker (albeit it being skew). Was there a significant difference between these bands to enable their clear distinction? Or were they selected because they were expected? Similarly, the polh band is marked in both BacPh and BacPhEn3 S2 and S3, when a similar sized band is present in each S4 lane. The decreased band intensity could be due to loss of the polh protein through the purification steps and should not be an eliminating factor. What enabled the distinction of the polh band in BacPh S3 from the band of similar size in S4? While I don't doubt the bands indicated are the target bands, I am cautious about the process used to determine these results, particularly when presented against the backdrop of other host and viral proteins. 

My  second comment relates to the dose response assays. I find the results very interesting and would like to comment on the need to presume the presence of the En3 (line 370). The authors indicate that BacPhEn3 S4 contains the En3 protein, but does not have polh protein as per figure 4 (indicating a lack of OBs). This presents an opportunity to evaluate the En3 proteins affect on the BacPh OBs via a series of combinations. Administering doses of BacPhEn3 S4 alone to larvae should not result in significant larval mortality due to the absence of OBs (unless En3 can cause serious harm alone which would itself be interesting or if there are somehow still sufficient OBs in this sample). However, when mixing BacPhEn3 S4 with the BacPH OBs (or any other virus), one might expect similar levels of mortality to that of the BacPhEn3 OBs + cell lysate while also expecting a greater effect than the BacPh OBs alone. These would form some interesting hypothesis which could be tested while also providing further evidence that the improvements are due to the presence of En3. It would also support the presence of En3 in the supernatant (soluble phase) and the findings of this study.

Minor edits:

L195: Sample 5 should be abbreviated S5, not P5.

L248: Fig 1b appears to be an agarose gel, but the colours have been inverted making it look more akin to an SDS-PAGE gel such as in Fig 4. Please clarify if this is an inverted image of an agarose or PAGE gel in text or revert to the standard format for agarose gels such as in Fig 1a and Fig 3.

L304: The use of asterisks to indicate three different proteins is not ideal. For example, BacPhEn3 has two asterisks one for polh and another for En3. This requires the reader to go through a process of elimination to determine what each asterisks marks rather than determining this immediately.

L360-L379: I assume the results presented here are from the dose response assay, however, the way it is written gives the impression that the LC50 values were calculated from the discriminating concentration assay reported in the previous paragraph. An indication that these are results from the dose response assay in L360-362 would easily remove any miscomprehension.

L425: The word among appears to be accidentally repeated "among the different among species".

L445: The word clearly appears to be accidentally repeated "clearly have clearly different".

Appendix B: What does the red text indicate?

Kind regards

Author Response

Reviewer 1 Major Comments

Author reply

My first comment relates to Figure 4 and how the various proteins were identified. It would appear bands were selected based on estimated sizes of the target proteins rather than via any form of purification or specific detection (such as western blot). Was there some form of analysis not mentioned in the methods section which aided in the identification of proteins?

The reviewer is correct. The different proteins were identified based on the estimated sizes of the target proteins. We did not performed a Western blot as the protein of interest, in this case the enhancin, was not marked with a His-tag. We recognize that we could have ordered a specific antibody to overcome this issue. We avoided using a His-tag for protein identification and purification as N-terminal His-tags reduce the thermal stability of the proteins which affects both protein function and binding properties (Booth et al., 2018). Most importantly, His-tagging of Bacillus enhancin-like (Bel) proteins resulted in the loss of mucin degradation activity (Galloway et al., 2005; Wang and Granados, 1997). As the enzymatic activity of the enhancin was critical to the objectives of our study we were obliged to avoid the use of His-tags and did not perform a Western. We have now mentioned this issue in the Materials and Methods section (lines 97-100).

For example, the authors mark two bands in S2 and S4 of BacPhEn3 as enhancin, however, a similar sized protein band is visible in BacPh S2 when roughly based on the marker (albeit it being skew). Was there a significant difference between these bands to enable their clear distinction? Or were they selected because they were expected?

To improve clarity we have now labeled the bands of interest with letters (a,b,c) in Fig. 4b and have described their appearance and migration behavior in the text. Additional bands have been identified by a cross and asterisk (†, *) and described in the text.

The enhancin band was identified based on its expected size. This has now been clarified in the Results section (lines 329-334).

Similarly, the polh band is marked in both BacPh and BacPhEn3 S2 and S3, when a similar sized band is present in each S4 lane. The decreased band intensity could be due to loss of the polh protein through the purification steps and should not be an eliminating factor. What enabled the distinction of the polh band in BacPh S3 from the band of similar size in S4? While I don't doubt the bands indicated are the target bands, I am cautious about the process used to determine these results, particularly when presented against the backdrop of other host and viral proteins.

In S4 BacPh and BacPhEn3 a band of similar size to Polh was present (now marked with a cross) but it was clearly of lower intensity than in S2 and S3. We did not confirm that this was the Polh band. Due to the sample preparation procedure the S4 sample may have contained traces of polyhedrin (now discussed in the text), or this band may be another cellular protein as it also appeared in S1 (BV fraction).  We have now explained this in detail in the Results section (lines 346-352).

My second comment relates to the dose response assays. I find the results very interesting and would like to comment on the need to presume the presence of the En3 (line 370). The authors indicate that BacPhEn3 S4 contains the En3 protein, but does not have polh protein as per figure 4 (indicating a lack of OBs). This presents an opportunity to evaluate the En3 proteins affect on the BacPh OBs via a series of combinations. Administering doses of BacPhEn3 S4 alone to larvae should not result in significant larval mortality due to the absence of OBs (unless En3 can cause serious harm alone which would itself be interesting or if there are somehow still sufficient OBs in this sample). However, when mixing BacPhEn3 S4 with the BacPH OBs (or any other virus), one might expect similar levels of mortality to that of the BacPhEn3 OBs + cell lysate while also expecting a greater effect than the BacPh OBs alone. These would form some interesting hypothesis which could be tested while also providing further evidence that the improvements are due to the presence of En3. It would also support the presence of En3 in the supernatant (soluble phase) and the findings of this study.

We agree. The assays proposed by reviewer 1 are interesting and right now we are conducting a series of assays with the soluble fraction of the cells (BacPhEn3 S4) in mixtures with OBs of a range of other baculovirus species in different lepidopteran hosts. The results of that study will form part of another paper that we will publish separately.  We appreciate the reviewer's suggestion.

Reviewer 1 Minor edits

Author reply

L195: Sample 5 should be abbreviated S5, not P5.

Done (line 204).

L248: Fig 1b appears to be an agarose gel, but the colours have been inverted making it look more akin to an SDS-PAGE gel such as in Fig 4. Please clarify if this is an inverted image of an agarose or PAGE gel in text or revert to the standard format for agarose gels such as in Fig 1a and Fig 3.

This is an image of the agarose gel of PCR products obtained using M13-Fw and M13-Rv primers that has been inverted to improve readability. As requested, we have now clarified this in the figure caption (lines 259-260).

L304: The use of asterisks to indicate three different proteins is not ideal. For example, BacPhEn3 has two asterisks one for polh and another for En3. This requires the reader to go through a process of elimination to determine what each asterisks marks rather than determining this immediately.

We appreciate this observation. We have now indicated each protein in the SDS-PAGE with a letter that corresponds to each protein to avoid confusion. We also indicated two bands that are mentioned in the text with symbols (cross and asterisk). We have modified Fig. 4b and the figure legend accordingly (lines 318-322).

L360-L379: I assume the results presented here are from the dose response assay, however, the way it is written gives the impression that the LC50 values were calculated from the discriminating concentration assay reported in the previous paragraph. An indication that these are results from the dose response assay in L360-362 would easily remove any miscomprehension.

Sorry for the confusion. We have now clarified the text in the Results section (lines 384-389).

L425: The word among appears to be accidentally repeated "among the different among species".

Deleted (line 466).

L445: The word clearly appears to be accidentally repeated "clearly have clearly different".

Corrected (lines 486).

Appendix B: What does the red text indicate?

The red text indicates the transmembrane domains. We have now included this information in the figure caption (line 504) and in Appendix B (line 640)

Reviewer 2 Major points:

Author reply

I would suspect that non-occluded viruses were present within the infected cells and that this influenced the infectivity of the inoculum to a certain degree. The authors might discuss the assessment of this potential influence in the text.

The presence of non-occluded viruses in samples produced in cells cannot be discounted. However, if present, such entities would be present in both control (BacPh) and BacPhEn3 treatments involving cell lysate, so that we are confident that effects attributed to the presence of enhancin are real and not an inadvertent effect of contamination by non-occluded virions.  This is a useful point and we have consequently inserted this text in the Discussion (lines 431-437).

I would feel that the conduct of western blotting in addition to the SDS-PAGE would have led to more rigorous conclusion in searching for the enhancin in five fractions, although I do not necessarily require the experiment here.

The reviewer is correct in that identification of the enhancin band would have been strengthened by a Western blot. The different proteins were identified based on the estimated sizes of the target proteins. We did not performed a Western blot as the protein of interest, in this case the enhancin, was not marked with a His-tag. We recognize that we could have ordered a specific antibody to overcome this issue. We avoided using a His-tag for protein identification and purification as N-terminal His-tags reduce the thermal stability of the proteins which affects both protein function and binding properties (Booth et al., 2018). Most importantly, His-tagging of Bacillus enhancin-like (Bel) proteins resulted in the loss of mucin degradation activity (Galloway et al., 2005; Wang and Granados, 1997). As the enzymatic activity of the enhancin was critical to the objectives of our study we were obliged to avoid the use of His-tags and did not perform a Western. We have now mentioned this issue in the Materials and Methods section (lines 97-100).

Reviewer 2 Minor points:

Author reply

Abstract, line 15: The sentence “they facilitate the fusion of ODVs to midgut cells.” should be revised or deleted because this possible action of alphabaculovirus enhancin has not been supported by strong evidence until now.

Removed (line 15).

Introduction, lines 58 and 417: The authors mentioned the protein GP37, which is baculoviral fusolin. The authors should briefly explain fusolin, a viral enhancing factor other than enhancin, by citing a study that fusolin derived from an entomopoxvirus was expressed using a bacmid and its strong ability to enhance an NPV was verified (J. Virol. 2008, 82 (24), 12406–12415.).

Thank you for pointing out this reference. We have now made mention of this fusolin in the Introduction (lines 59-60) and the Discussion (lines 457-459).

Lines 184 and 190: Please describe gravity instead of “rpm”.

Done (lines 159, 168, 182, 194, 199).

Line 194: Please give more details (pore size of the filter, gravity and time of the centrifugation).

Done (lines 203 and 204).

Line 522: “215” should be “2015”.

Done (line 564).

Reviewer 2 Report

This manuscript describes that an enhancin derived from a betabaculovirus, one of viral enhancing factors, accumulated within infected cells but not within alphabaculovirus OBs nor in its ODVs. The localization in infected cells was reported one time but has not been verified. This manuscript suggested based on several experiments that the infected cells were the third place where the enhancin accumulated when it was expressed under certain conditions. This is a new characteristic of enhancin. Enhancin is a potential enhancing agent for microbial insecticides such as insect viruses and Bacillus thuringiensis. Therefore, this manuscript will contribute to science of biological control concerning enhancing factors.

Major points:

I would suspect that non-occluded viruses were present within the infected cells and that this influenced the infectivity of the inoculum to a certain degree. The authors might discuss the assessment of this potential influence in the text.

I would feel that the conduct of western blotting in addition to the SDS-PAGE would have led to more rigorous conclusion in searching for the enhancin in five fractions, although I do not necessarily require the experiment here. 

Minor points:

Abstract, line 15: The sentence “they facilitate the fusion of ODVs to midgut cells.” should be revised or deleted because this possible action of alphabaculovirus enhancin has not been supported by strong evidence until now.

Introduction, lines 58 and 417: The authors mentioned the protein GP37, which is baculoviral fusolin. The authors should briefly explain fusolin, a viral enhancing factor other than enhancin, by citing a study that fusolin derived from an entomopoxvirus was expressed using a bacmid and its strong ability to enhance an NPV was verified (J. Virol. 2008, 82 (24), 12406–12415.).

Lines 184 and 190: Please describe gravity instead of “rpm”.

Line 194: Please give more details (pore size of the filter, gravity and time of the centrifugation).

Line 522: “215” should be “2015”.

Author Response

(The authors gave the same response as above.)
